# Database size positively correlates with the loss of species-level taxonomic resolution for the 16S rRNA and other prokaryotic marker genes

Seth Commichaux[1]*, Tu Luan[2,3], Harihara Subrahmaniam Muralidharan[2,3], Mihai Pop[2,3]

1 Center for Food Safety and Nutrition, Food and Drug Administration, Laurel, Maryland, United States of America, 2 Department of Computer Science, University of Maryland, College Park, Maryland, United States of America, 3 Center for Bioinformatics and Computational Biology, University of Maryland, College Park, Maryland, United States of America

☉ These authors contributed equally to this work.
* Seth.Commichaux@fda.hhs.gov

**Data Availability Statement:** All the data used for this study is publicly available. The code used for our analysis, and data used to plot the figures, can be found at https://github.com/tluan/

## Abstract

For decades, the 16S rRNA gene has been used to taxonomically classify prokaryotic species and to taxonomically profile microbial communities. However, the 16S rRNA gene has been criticized for being too conserved to differentiate between distinct species. We argue that the inability to differentiate between species is not a unique feature of the 16S rRNA gene. Rather, we observe the gradual loss of species-level resolution for other nearly-universal prokaryotic marker genes as the number of gene sequences increases in reference databases. This trend was strongly correlated with how represented a taxonomic group was in the database and indicates that, at the gene-level, the boundaries between many species might be fuzzy. Through our study, we argue that any approach that relies on a single marker to distinguish bacterial taxa is fraught even if some markers appear to be discriminative in current databases.

## Author summary

The use of reference databases for assigning taxonomic labels to genomic and metagenomic sequences is a fundamental bioinformatic task in the characterization of microbial communities. The increasing accessibility of high throughput sequencing has led to a rapid increase in the size and number of sequences in databases. This has been beneficial for improving our understanding of the global microbial genetic diversity. However, there is evidence that as the microbial diversity is more densely sampled, increasingly longer genomic segments are needed to differentiate between distinct bacterial species. The scientific community needs to be aware of this issue and needs to develop methods that better account for it when assigning taxonomic labels to metagenomic sequences from microbial communities.

ProkaryoticMarkerGenes-DatabaseSizeAnalysis.
The SILVA database used can be downloaded from
https://www.arb-silva.de/fileadmin/silva_
databases/release_138.1/Exports/SILVA_138.1_
SSURef_tax_silva.fasta.gz. The GTDB marker
genes (release version 207) can be downloaded
from https://data.gtdb.ecogenomic.org/releases/
release207/207.0/genomic_files_all/bac120_
marker_genes_all_r207.tar.gz. The NCBI Assembly
database accessions for the Listeria assemblies
can be found in S1 File.

**Funding:** TL, HSM, and MP were funded by the
National Institutes of Health [R01-AI-100947]. The
funders had no role in study design, data
collection, and analysis, decision to publish, or
preparation of the manuscript.

**Competing interests:** The authors have declared
that no competing interests exist.

## Introduction

The 16S rRNA gene has been employed for decades to taxonomically characterize the prokaryotes found in microbial communities. Several databases have been created as a reference for assigning taxonomic labels to newly sequenced versions of the 16S rRNA gene. For example, the RDP [1], Green Genes [2], and SILVA [3] database comprise almost 4 million distinct gene variants. However, although the 16S rRNA gene is universally present in the genomes of prokaryotes, and is phylogenetically-informative (i.e., useful for inferring their evolutionary history), its use to taxonomically-characterize microbial communities has been criticized due to its limited taxonomic resolution for some taxa [4,5].

With the advent of metagenomics—the culture-independent sequencing and analysis of the total organismal DNA directly extracted from a sample—a broader range of methods have been developed to characterize the taxonomy of microbial communities. Like the 16S rRNA, metagenomic methods use reference databases to assign taxonomic labels to sequences. However, instead of amplicon sequence variants (ASVs) or operational taxonomic units (OTUS), in a metagenomic context the database might consist of k-mers, genes, or genomes, and the sequences being taxonomically classified might be metagenomic reads, contigs, or metagenome-assembled-genomes (MAGs). While, ideally, classification would rely on a holistic interpretation of genomic data, many metagenomic tools that use gene databases still rely on information from the 16S rRNA gene or sets of universal single-copy genes considered in isolation from each other [6–13]. Gene catalogs, composed of a non-redundant collection of the genes extracted from the genomes and metagenomes collected from a specific habitat (e.g., the human gut), have also been widely used to assign taxonomic labels, frequently considering each gene independently [14–23]. In other words, even in cases when a method may, at first glance, appear to consider a broader genomic context than any individual gene, the actual classification frequently still relies on just one gene or small genomic region considered in isolation.

Here we propose that the limited resolution of the 16S rRNA gene as a taxonomic marker reflects a more fundamental challenge–the impact of growing reference databases on the accuracy of taxonomic classification approaches that rely on a single genomic marker. We hypothesize that the sequence diversity of most taxa is highly under-sampled, thereby artificially inflating the discriminative power of individual genes, but as databases accumulate an increasing number of sequences from diverse species, the likelihood of encountering interspecies sequence collisions rises.

Interspecies sequence collisions within databases reduce the amount of information for differentiating the species in a database and have various well-known consequences for downstream analyses. For example, metagenomic reads can multi-map, i.e., align equally well to multiple sequences in the database, affecting the accuracy of abundance estimation and taxonomic classification [24]. Some tools for metagenomic analysis explicitly handle multi-mapped reads, e.g., by filtering out alignments based upon coverage statistics or by assigning the taxonomy of the lowest common ancestor (LCA) of the mapped database sequences [12,25]. Other methods try to handle sequence collisions at the level of the database, e.g., by first clustering the genes within the database as in the case of the construction of gene catalogs. However, previous work has shown that even after gene clustering, up to 40% of metagenomic reads can multi-map within a gene catalog. Further, gene clustering can lead to a loss of species representation in catalogs [24].

Here we demonstrate that the limited resolution of the 16S rRNA gene as a taxonomic marker reflects a more fundamental limitation of performing taxonomic classification based on any individual taxonomic maker. As sequence databases increase in size, and as under-

sampled taxa become better represented in databases, the resolution of taxonomic classifications necessarily degrades due to an increase in interspecies sequence collisions. This effect has previously been demonstrated for k-mer databases [26]. Here, we demonstrate the same pattern when analyzing marker gene sequences. The sequences analyzed include the 16S rRNA gene and single copy marker genes that are phylogenetically informative and nearly universally present in prokaryotes [4,27], including a set of 40 single copy marker genes [28], and the 120 genes used by the Genome Taxonomy Database (GTDB) [29].

## Methods

### Sequence data used for analysis

The SILVA 16S rRNA gene database (release 138.1) was downloaded and filtered to remove sequences with incomplete taxonomic labels and those from mitochondria and plastids (394,617 sequences remained). We also downloaded the 120 marker genes (35,171,383 sequences) from the Genome Taxonomy Database (GTDB) (release 207). It should be noted that both databases contain full-length genes that have not been deduplicated, i.e., there can be identical sequences from the same species.

To create the *Listeria* gene databases we first downloaded the 5,014 RefSeq genome sequences available for *Listeria* (representing 34 distinct species) in the NCBI Assembly database. Most genome sequences from this genus (4,439) belonged to *Listeria monocytogenes*. From each *Listeria* genome sequence we extracted the 16S rRNA gene using Barrnap [30] (7,625 sequences) and 40 prokaryotic marker genes using fetchMG [31]. The output of fetchMG was filtered, for each marker gene, by removing sequences that were of very different length (below half or above twice as long as the average sequence) and, thus, likely to be artifacts.

To assess the overlap between the 120 marker genes used by the GTDB and the 40 universal marker genes previously used for taxonomic classification (e.g., by the mOTU tool [13]), we ran fetchMG on the GTDB gene set. We identified 29 marker genes that were shared between the two data sets.

### Database simulation and clustering

To create the simulated databases for each marker gene in the SILVA and GTDB, we created a collection of random subsets varying in size from 10,000 to 200,000 sequences in 10,000 gene increments. For the *Listeria* marker genes, we randomly subsampled its sequences into subsets varying in size from 1,000 to 5,000 sequences in 1,000 gene increments. We repeated this process 100 times so we could estimate the variability of our results.

Each simulated database was clustered with CD-Hit [32] at several sequence identity cutoffs (95%, 97%, 99%, 100%), requiring that shorter sequences fully align to longer ones. It is important to note that we clustered the database sequences rather than the query sequences (as would normally be done if we constructed ASVs or OTUs from amplicon or metagenomic data). The clustering thresholds were chosen for several reasons. Firstly, genes sharing high sequence similarity are likely to contain regions where metagenomic reads cannot be uniquely aligned, thus, highlighting issues with read-based taxonomic classification. Further, genes from distinct species that are 100% identical indicate that species-level resolution is simply not possible for a specific gene. Secondly, these thresholds are often applied in various bioinformatic contexts. For example, microbial gene catalogs are often clustered at 95% identity to produce species-level gene clusters [24], while 97% and 99% identity have been used as proxy species-level thresholds in OTU clustering [33].

## Cluster analysis

To assess the level of ambiguity present in a database in a classifier-independent manner, we clustered the database sequences at different identity cut-offs and computed the number of clusters that contained sequences from more than one species (multi-species clusters). Multi-species clusters approximate the likelihood that a classifier would be unable to distinguish between the distinct species found in the cluster. At the 100% threshold, for example, a multi-species cluster implies that identical sequences have been assigned divergent taxonomic labels, thus no sequence feature would be able to distinguish the co-clustered sequences. The other thresholds we use are frequently used in the bioinformatics community to define boundaries between taxonomic groups, under the assumption that sequences that have a higher level of similarity have the same taxonomic label. For example, microbial gene catalogs are often clustered at 95% identity to produce species-level gene clusters [24], while 97% and 99% identity have been used as species-level thresholds in OTU clustering [33].

The rate at which sequences were clustered with sequences from other species was estimated using the $Y = cX^m$ linear regression model in log-log space, where $m$ is the rate, $Y$ is the number of sequences in multi-species clusters, and $X$ is the number of genes in the simulated database.

## Results

### Analysis of the databases simulated from the SILVA database and GTDB

To explore the extent to which database growth impacts the specificity of taxonomic labels in a broad context we analyzed random subsamples of the SILVA database and the GTDB (Fig 1). For each marker gene (the 16S rRNA and the 120 marker genes of the GTDB), the number of sequences in multi-species clusters increased at a super-linear rate as the database grew (Fig 2A and 2B), with higher rates of growth when clustered at higher similarity thresholds.

The fraction of species that had at least one sequence in a multi-species cluster increased with database size (Fig 2C); albeit this fraction was lower when clustering with higher similarity thresholds. Similarly, the percentage of species that belonged to multi-species clusters for at least 50 of the 120 GTDB marker genes increased with database size (S1 Fig), suggesting that even methods that can effectively integrate data from multiple genes may be affected by database growth. The number of multi-species clusters depended on the density at which a particular taxonomic group was sampled, with a positive linear correlation between the number of distinct species within a genus and the number of multi-species clusters formed by sequences from that genus (Fig 2D). Importantly, amongst the species pairs that most frequently cooccurred in multi-species clusters when clustered at 100% identity were pathogens like *Bacillus anthracis* and *Vibrio cholerae*, which clustered with non-pathogenic species from their respective genera (S1 Table).

### Analysis of the *Listeria* marker gene databases

To explore the extent to which database growth impacts the specificity of taxonomic labels in a context where the labels have a potential health implication, we focused on a single deeply-sampled genus, *Listeria*, which includes the important foodborne pathogen *Listeria monocytogenes* (Fig 3). The NCBI Assembly database contained 5,014 RefSeq genome sequences from 34 distinct species of *Listeria*. Most of the assemblies (4,439) belonged to *Listeria monocytogenes*. In these assemblies we identified 7,625 16S rRNA gene sequences (ranging from 1 to 9 copies per genome, consistent with the expectation that this gene is often multicopy in *Listeria*)

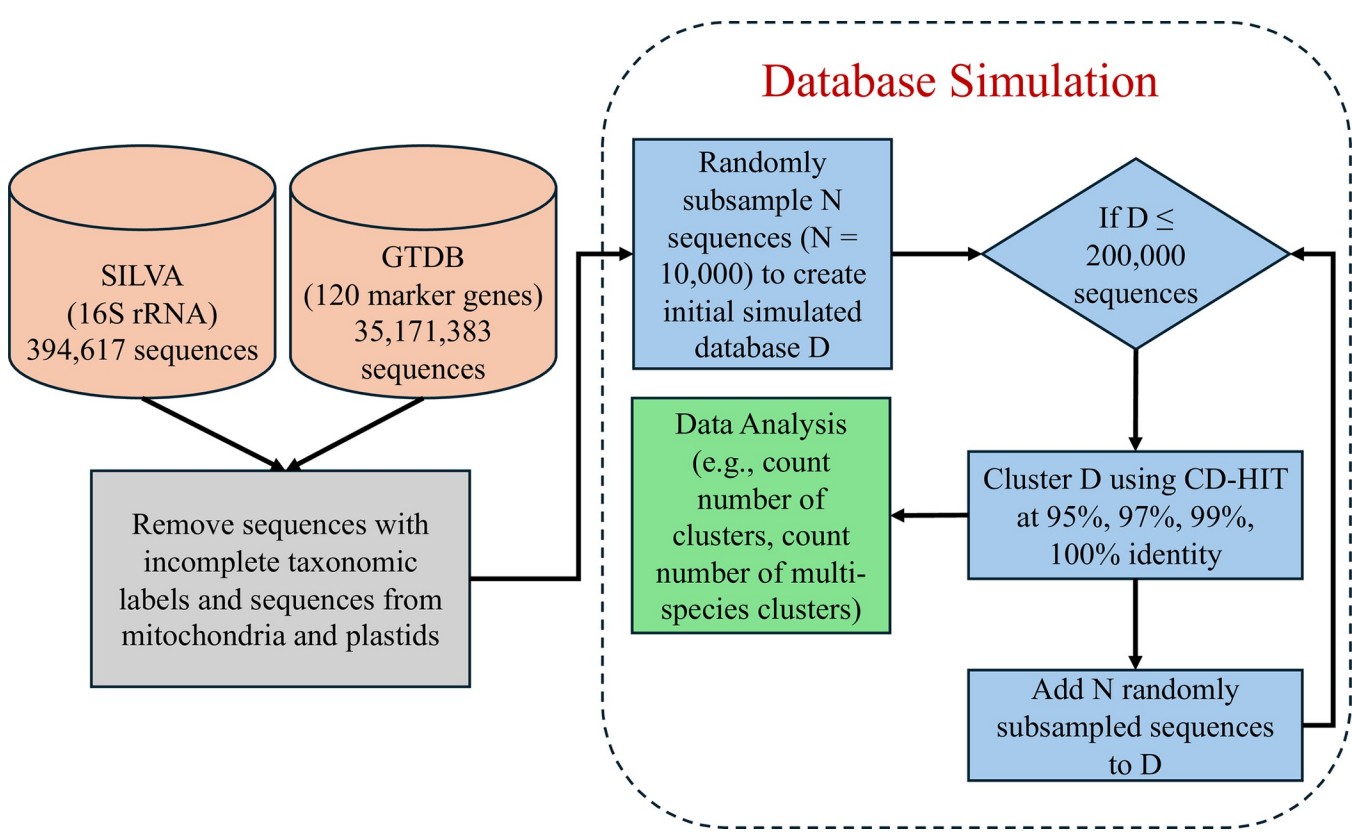

**Fig 1. Workflow diagram of the analysis done for the SILVA database and the GTDB.** The SILVA and GTDB were downloaded and sequences with incomplete taxonomic labels or from mitochondria and plastids were removed. To create the simulated databases for each marker gene, we created a collection of random subsets varying in size from 10,000 to 200,000 sequences in 10,000 gene increments. Each simulated database was clustered at 95%, 97%, 99%, and 100% identity requiring that shorter sequences fully align to longer ones.

and 200,359 marker gene sequences (~40 per genome, consistent with the expectation that these genes are mostly single copy).

As observed in the previous experiments, even within this single genus, the number of sequences in multi-species clusters increased with the database size at all clustering thresholds and for all marker genes, including the 16S rRNA gene (Fig 4). Notably, at 95% identity each *Listeria* species had 16S rRNA gene sequences in multi-species clusters when analyzing the full data set. Further, 25% (100% identity) to 65% (95% identity) of all the *L. monocytogenes* sequences occurred in multi-species clusters.

## Discussion

Our results support the observation that as sequence databases grow to contain more species and sequences, they are progressively losing species-level resolution for taxonomic classification [26]. This trend was strongly correlated with how well-represented a taxonomic group was in the database and indicates that, at the gene-level, the boundaries between many species might be fuzzy. The main difference between marker genes was the rate at which they lost species-level resolution, with the 16S rRNA gene sometimes being an outlier (a previously noted trend [4,5]). Notably, at lower clustering thresholds (corresponding to longer evolutionary distances), the 16S rRNA gene was less informative than other marker genes, while at the highest

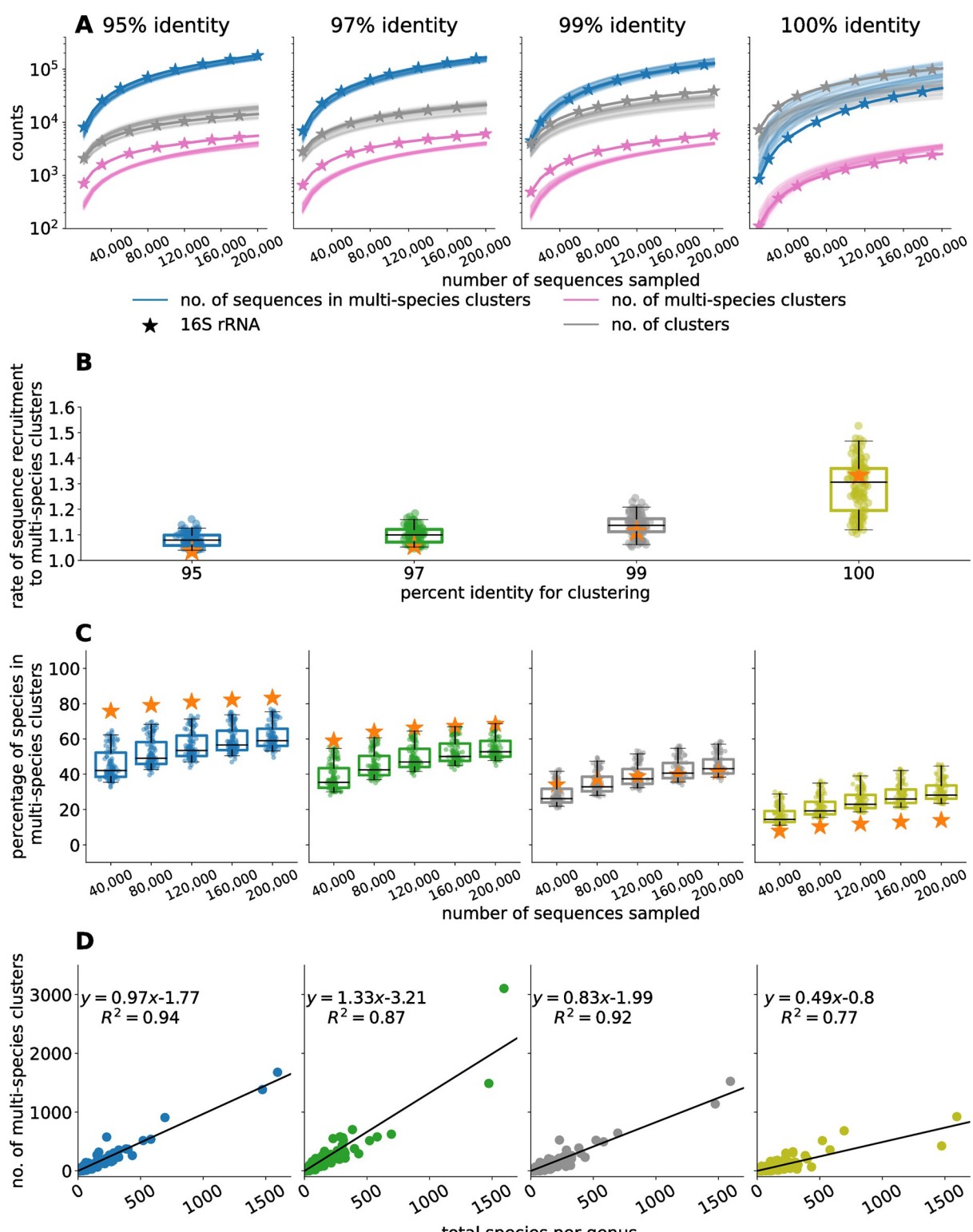

**Fig 2. Clustering analysis for simulated databases created by randomly sampling sequences from the 16S rRNA SILVA database and the 120 marker gene Genome Taxonomy Database (GTDB).** Each simulated database was clustered at 95%, 97%, 99%, and 100% identity requiring that shorter sequences fully align to longer ones. The 16S rRNA gene is denoted by a star in all subplots. A) The relationship between the number of genes in the simulated databases, the number of clusters, the number of multi-species clusters, and the number of sequences in multi-species clusters. For GTDB, each curve is for one of the 120 marker genes. B) The rate at which sequences were recruited to multi-

species clusters as the database grows. Each point represents one of the 120 marker genes in the GTDB. C) The percentage of species with sequences in multi-species clusters. D) The relationship between the number of multi-species clusters that a species belongs to and the species richness of its genus (i.e., the total number of species from that genus) in the simulated database. This data was only taken from the final iteration of the simulated databases. The results were aggregated across all 120 marker genes in the GTDB.

threshold of 100% identity the 16S rRNA gene showed higher discrimination, likely due to the higher level of short-term sequence evolution within hypervariable regions of this gene.

Our observations have broad implications given that genes are routinely employed for various bioinformatic tasks, such as the taxonomic classification of genomic and metagenomic reads and assemblies, the curation of gene and metagenome-assembled-genome (MAG) catalogs, phylogeny estimation, and abundance profiling [3,24,29,34,35]. The progressive loss of species-level resolution for taxonomic classification calls into question highly specific taxonomic classifications (e.g., species, strain) for individual metagenomic reads and genes that are simplistically assigned based upon sequence similarity to database sequences. While, at some level, the community understands that accurate classification requires the use of whole-genome data [29,36,37], many computational tools for metagenomic analysis initially classify

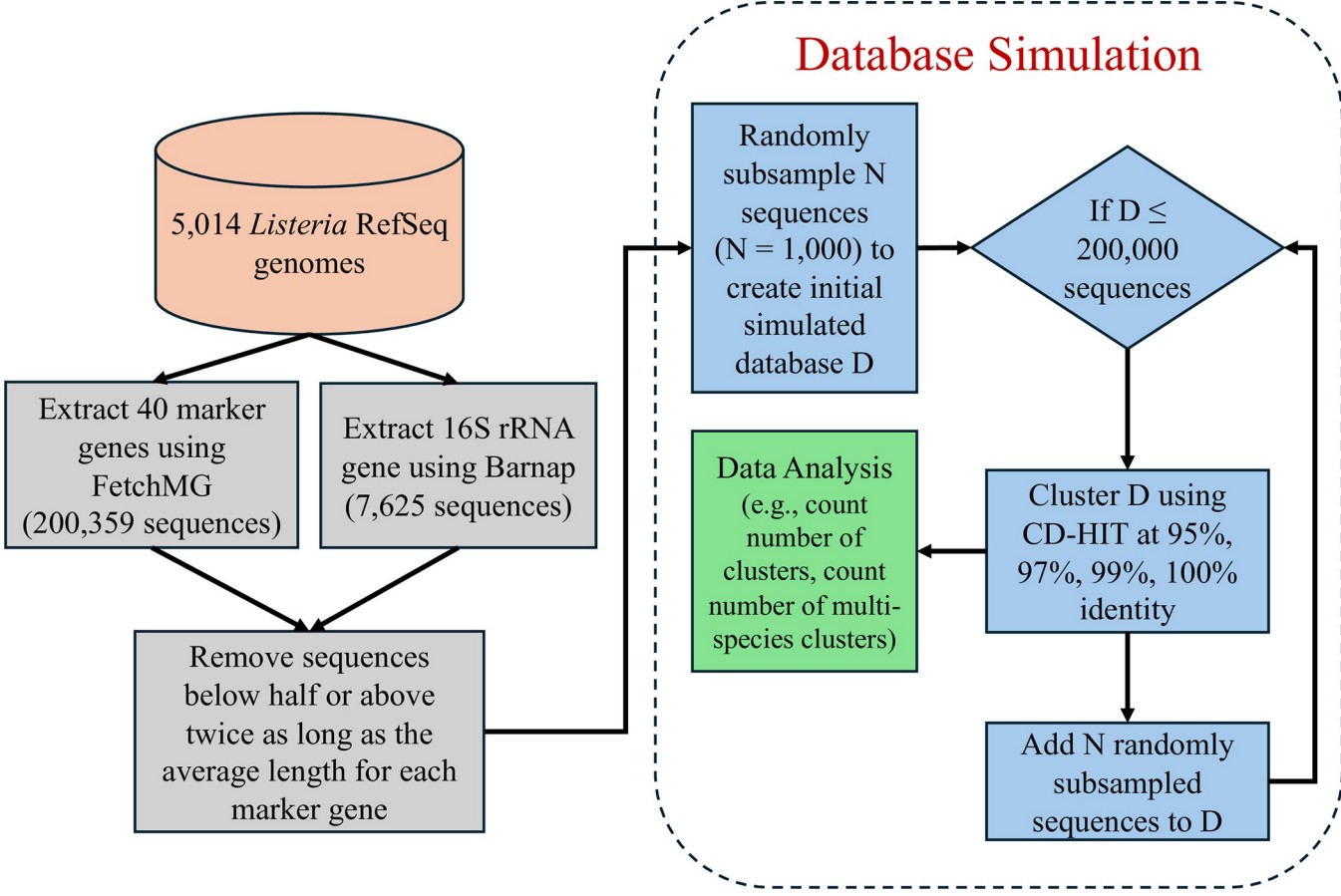

**Fig 3. Workflow diagram of the analysis done for the *Listeria* marker gene simulated databases (16S rRNA and 40 marker genes).** First, 5,014 *Listeria* draft genomes were downloaded from RefSeq and the 16S rRNA and 40 markers genes were predicted with Barnap and FetchMG, respectively. Genes that were below half or above twice as long as the mean length for a specific marker gene were removed. To create the simulated databases for each marker gene, we randomly subsampled the sequences into subsets varying in size from 1,000 to 5,000 sequences in 1,000 gene increments. We repeated this process 100 times so we could estimate the variability of our results. Each simulated database was clustered at 95%, 97%, 99%, and 100% identity requiring that shorter sequences fully align to longer ones.

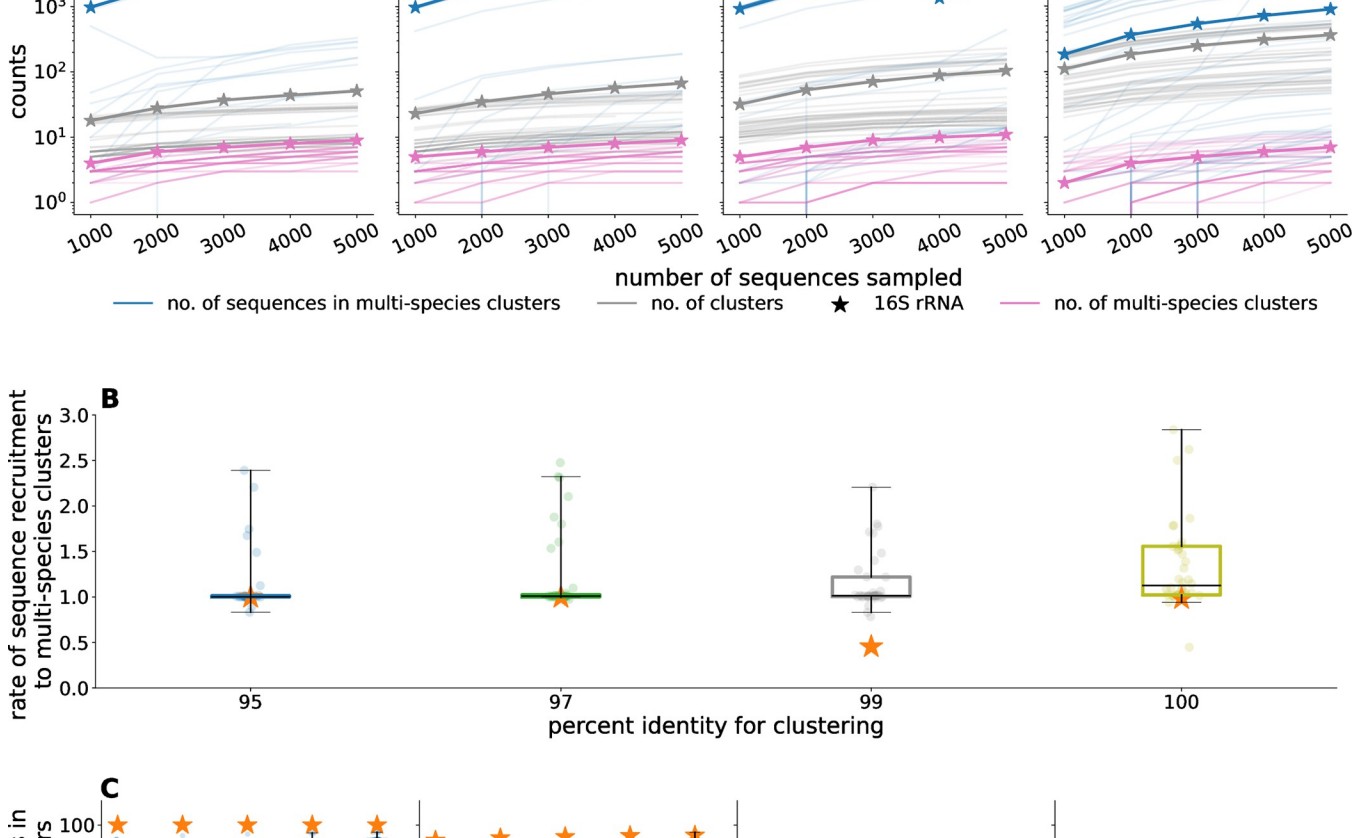

**Fig 4. Clustering analysis for the simulated databases created by randomly sampling sequences from the 16S rRNA and the 40 marker genes extracted from 5,014 *Listeria* genomes.** Each simulated database was clustered at 95%, 97%, 99%, and 100% identity requiring that shorter sequences fully align to longer ones. The results for each gene are reported by the median over 100 bootstrap experiments. The 16S rRNA gene is denoted by a star in all subplots. A) The relationship between the number of genes in the simulated databases, the number of clusters, the number of multi-species clusters, and the number of sequences in multi-species clusters. Each curve represents one of the 40 marker genes. The starred curve represents the 16S rRNA gene B) The rate at which sequences were recruited to multi-species clusters as the database grows. Each point represents one of the 40 marker genes. C) The percentage of species with sequences in multi-species clusters.

individual k-mers, reads, and genes, even if they later aggregate the results [1,6–13,35,38,39]. Our results indicate that such methods might provide overly specific taxonomic assignments for taxa that are underrepresented in the reference database used for analysis. Further, our results reemphasize the issues associated with the methods used to create microbial gene catalogs, which typically cluster taxonomically unlabeled genes into a set of representative

sequences, which are then assigned taxonomic labels. Such workflows have been shown to create multi-species clusters, irrespective of the sequence identity threshold used for clustering, thereby effectively erasing entire species from the catalogs [24].

We suggest that taxonomic classifiers account for how densely a particular taxonomic group is represented in a database when assigning taxonomic labels. Future work should continue to explore what fraction of a genome is required to accurately identify a particular species within metagenomic data, and to quantify the taxonomic information content of different genomic regions. Instructively, a recent study using a complete set of universally present prokaryotic marker genes (unlike this study's focus on individual genes, but including the sets we used) found that 97.7% of species could be differentiated, but only 8.8% of strains [40]. Further, even when comparing the average nucleotide identity (ANI) of the entire shared gene or genomic content of genomes, the commonly used threshold of 95% ANI is inconsistent as a proxy for species. For example, there are species within genera like *Brucella* or *Mycobacterium* that can only be differentiated above 99.5% ANI [37,41]. And in alignment with our findings, highly sampled species (defined in the study as species with over 100 sequenced genomes) were more likely to comprise strains that diverged by more than the 95% cutoff [42].

Another database issue that was not explored in our analysis but that is worth mentioning is the issue of mislabeled sequences in databases. Studies have demonstrated that taxonomic classification tools such as the RDP classifier can be sensitive to noise and that including even one mislabeled sequence during training can lead to a cascade of misclassifications [1,43]. This observation combined with the findings of this current study highlights the importance of careful data curation to diminish analytic errors.

At the broadest level, our work reiterates the long-noted issues with using discrete and definitive taxonomic labels for DNA sequences that are continually evolving [44,45]. Recombination alone can render a genome a mosaic of lineages, with some horizontally transferred sequences potentially unrelated to any specific lineage. And the implications extend beyond taxonomic analysis to analyses that rely on taxonomy as a proxy for other properties of organisms, such as the functional profile of microbial communities [46]. A better approach would combine the information from taxonomic and functional markers in an application-specific manner. For example, our analysis has shown that single-gene analyses are unable to discriminate important human pathogens from their near, non-pathogenic neighbors. This supports that the classification of pathogens is not a strict function of taxonomy (e.g., some strains of *E. coli* are beneficial gut microbes while others cause food poisoning) and suggests that pathogen classification should include genomic markers functionally associated with pathogenesis and virulence.

Altogether, it is important that future work continue exploring the relationship between molecular evolution, taxonomy, function, the composition of sequence databases, and the fidelity of annotations when using reference databases as substrates for metagenomic analysis.

## Supporting information

**S1 Fig. The percentage of species that belong to multi-species clusters for at least 50 of the 120 GTDB marker genes.** The percentage of species in multi-species clusters is also plotted for the 16S rRNA gene (denoted by a star in all subplots). The simulated databases were created by randomly sampling sequences from the 16S rRNA SILVA database and from the 120 genes used by the Genome Taxonomy Database (GTDB). Each simulated database was clustered with CD-Hit at several sequence identity cut-offs (95%, 97%, 99%, 100%), requiring that shorter sequences fully align to longer ones.
(TIF)

**S1 File. A list of the NCBI accessions for the RefSeq assemblies used for the *Listeria* analysis.**
(TXT)

**S1 Table. A table showing the number of times species-pairs occur in multi-species clusters.** From the analysis of the SILVA database and the GTDB.
(XLSX)

## Author Contributions

**Conceptualization:** Seth Commichaux, Tu Luan, Harihara Subrahmaniam Muralidharan, Mihai Pop.

**Data curation:** Seth Commichaux, Tu Luan, Harihara Subrahmaniam Muralidharan, Mihai Pop.

**Formal analysis:** Seth Commichaux, Tu Luan, Harihara Subrahmaniam Muralidharan, Mihai Pop.

**Funding acquisition:** Mihai Pop.

**Investigation:** Seth Commichaux, Tu Luan, Harihara Subrahmaniam Muralidharan, Mihai Pop.

**Methodology:** Seth Commichaux, Tu Luan, Harihara Subrahmaniam Muralidharan, Mihai Pop.

**Project administration:** Seth Commichaux, Tu Luan, Harihara Subrahmaniam Muralidharan, Mihai Pop.

**Supervision:** Mihai Pop.

**Validation:** Seth Commichaux, Tu Luan, Harihara Subrahmaniam Muralidharan, Mihai Pop.

**Visualization:** Seth Commichaux, Tu Luan, Harihara Subrahmaniam Muralidharan, Mihai Pop.

**Writing – original draft:** Seth Commichaux, Tu Luan, Harihara Subrahmaniam Muralidharan, Mihai Pop.

**Writing – review & editing:** Seth Commichaux, Tu Luan, Harihara Subrahmaniam Muralidharan, Mihai Pop.

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
