## [Decision Letter · Decision Letter 0]

18 Mar 2024

Dear Dr. Commichaux,

Thank you very much for submitting your manuscript "Database size positively correlates with the loss of species-level taxonomic resolution for the 16S rRNA and other prokaryotic marker genes" for consideration at PLOS Computational Biology.

As with all papers reviewed by the journal, your manuscript was reviewed by members of the editorial board and by several independent reviewers. In light of the reviews (below this email), we would like to invite the resubmission of a significantly-revised version that takes into account the reviewers' comments.

We cannot make any decision about publication until we have seen the revised manuscript and your response to the reviewers' comments. Your revised manuscript is also likely to be sent to reviewers for further evaluation.

Sincerely,

Andre Kahles, PhD

Guest Editor

PLOS Computational Biology

Ilya Ioshikhes

Section Editor

PLOS Computational Biology

Reviewer's Responses to Questions

**Comments to the Authors:**

Reviewer #1: Summary

The study of microbial communities relies heavily on bioinformatic tools that can identify taxonomic composition of a metagenomic sample. This classification originally relied on 16S rRNA genes. However, with increased availability of shotgut metagenomic data, many current taxonomic classifiers rely on different sets of marker genes to determine taxonomy. One of the major advantages of using marker genes over 16S rRNA for taxonomic classification is the increased (species-level) resolution. All of the taxonomic classifiers rely on a reference database for their assignments; these databases are not static but are constantly growing. How the changes in the databases affect the accuracy of the classifiers is an important question that has not been thoroughly addressed in the field. In their paper, the authors claim that with increased sizes of genomic databases, i.e. with increased numbers of marker gene sequences, the marker genes used for taxonomic classification lose species-level resolution. The authors analyse different sets of marker genes, and for each marker gene, created a number of different sequence subsets of different sizes. Each of the subsets is then clustered at different identity cut-offs, and number of clusters containing sequences from multiple species is quantified. The authors noted that subsets with higher number of sequences resulted in higher number of clusters containing sequences from multiple species. However, this observation on its own is not particularly surprising (especially for 16S rRNA gene). It is also unclear whether this would have any effect on taxonomic classification. Importantly, while all the analyses in the paper are done on a per gene basis, the authors themselves note that taxonomic profiling at high resolution requires a combination of genomic markers. Indeed, current established taxonomic profilers (MetaPhlAn, GTDB, mOTUs) rely on combinations of marker genes for accurate taxonomic classification. Thus it remains unclear whether increase in numbers of sequences would actually result in less accurate taxonomic classification.

Major comments.

- The authors did not demonstrate that database size is a problem to existing taxonomic classifiers based on marker genes. While it would be an important task to test how database size affects accuracy of taxonomic classification, this was not done in this paper. On line 79, the authors state that ‘Multi-species’ clusters approximate the likelihood that a classifier would be unable to distinguish between the distinct species found’, suggesting that this could be a proxy for how accurate taxonomic classification would be. However, the authors provide no evidence to support this. As mentioned above, no classifier relies on a single marker gene for classification, and thus analysis at per gene level does not address the question. In addition, this completely disregards the algorithms behind different classifiers that might or might not be affected differently by databases of different size. In order to show that database size affects taxonomic classification accuracy, the authors would have to create databases of varying sizes for a set of commonly used taxonomic classifiers (see above) and use a ‘ground truth’ dataset to evaluate classification accuracy. Otherwise, it appears that they are calling attention to a problem that does not yet exist.

- The metrics used in the study are hard to interpret. For example, for figures 1A and 2A it is hard to assess importance/prevalence of the multi-species cluster without knowing the total numbers of clusters, these graphs might be more informative as a proportion. Similarly, for figure 1C, since the tools do no rely on a single gene, % of species with at least one gene in multi-species cluster is not very informative. More informative could be % of species with all or at least majority of marker genes in multi-species clusters.

- I could not find a link to the code used for the analysis of the data

Minor Comments

- Line 69: The overlap should be more than 3 genes, it would be good to see how the mapping between GTDB and 40 marker genes was done.

- Diagram of the workflow and a separate Methods section would be useful to assess the methods

- Choice of Listeria for the second part of analysis is unclear

- Linear regressions in Figure 1D have 2 outliers/high leverage points (one of which seems to have been cut out of second panel), these should either be done in log space or removed.

- On line 150 the authors state that many tools rely on individual genes for classification at species level, I am not sure which ones they mean (the references 16 and 20 do not, and k-mer based tools are not relevant to the analysis performed). The authors’ conclusion is that a combination of genomic markers should be used (and that longer genomic segments are needed to differentiate between distinct species), is in fact what is done by GTDB and other tools relying on marker genes.

Reviewer #2: This manuscript studied the common question that marker genes, such as 16S RNA is not enough to differentiate prokaryotes.

Major questions

1. As this study pointed out, the previous marker genes, including 16S RNA genes don’t provide high-enough resolution for discriminate microbes at the species-level. Based on your analysis, are you able to identify new genes that can be served as marker genes for this classification purpose or some gene combination? Or maybe the concept of “marker gene” should be reconsidered?

2. The authors also mentioned that even 95% ANI may not serve as the species level cut-off. Please provide more details regarding the genome completeness of the study that you cited (ref 18, 21, 22) and >=100 genome sequences is not clear enough. ANI is quiet common idea for prokaryotes classification. I would suggest the author to perform some analysis of some random genus to better show and support your results.

3. There is another idea, called pangenome, where scientists tried to use core genes for species- or genus- level classification. Maybe incorporating this layer of information will better answer the question proposed in this study. You can refer to this data which may serve as a case study to support your research. https://escholarship.org/uc/item/79t1w9dx

Minor questions:

1. “For all genes, the number of sequences in multi-species clusters increased at a super-linear rate

as the database grew (Figures 1A and 1B),”

This sentence is not clear. “For all genes”, do you mean the 120 marker genes you used? How do you define the “multi-species clusters”?

2. “The rate at which sequences are clustered with sequences from other species was estimated using the Y = cXm 97 linear regression model in log-log space,” I think you can use similarity to describe the how close the two sequences is. This makes readers easier to follow.

The Y-axis label of each subplot in Figure-1 is too close to the next figure, hard to tell is the title of which figure.

The legend mentions that “the number of genes, the number of sequences in multi-species clusters, the total number of multi-species clusters…”, while in Figure 1, the X-axis label is “counts”, the Y-axis label is “number of sequences sampled”. The terminology you use in both legend and figures should be consistent. Otherwise, it is really hard to follow. Please check all figures and legend to address this issue.

Reviewer #3: Overall I agree with the overall message of this manuscript, believe that it is sufficiently supported by the evidence presented, think this manuscript is a solid contribution to the field, and support its publication. Thoughts below are mostly about interpretation and language, and are offered as suggestions.

It is not clear how well random sampling of these reference databases captures the longitudinal dynamics (i.e. database growth over the years) that it is intended to be simulating. For example, the super-linear increase in sequences in multi-species clusters is necessarily true when subsampling in this way (at least averaged over subsampling realizations) since sequences are added to clusters at a uniform rate, and the number of clusters that are multi-species can only increase with size of the subsample. But the real growth in time of databases may not be the same as subsampling (where all re-orderings are equal, i.e. database entries are exchangeable). For example, the discovery of a new set of species that are well-separated from others could lead to a decrease in the fraction of sequences in multi-species clusters, that is not possible in the subsampling realization of database growth. Other dynamics (re-namings, splitting previously named species into multiple species etc.) also lead to entries that are not exchangeable with those that occurred before that event.

L61-62; L107-109; L130-131: All three frame the observation that “database growth is affecting our ability to discriminate important pathogens from their near neighbors.” as a degradation in species (or pathogen) resolution. I’m not sure I agree with that framing. I agree that the likelihood of single named-species being output as the query within some %ID threshold from the reference database does decrease. But previously that was a mistake due to an incomplete reference database. The change in performance seems to me an improvement, not a degradation.

L88-89: “Further, genes from distinct species that are 100% identical indicate that species level resolution is simply not possible for a specific gene.” I don’t agree with this. It shows there is a counter-example for that gene in which it is not possible to discriminate between a pair of species. That is, species resolution is not ALWAYS possible for that specific gene. But it may still be that that species-level resolution is often possible, e.g. Fig 1C 100% panel – ~20% of species are in multi-species cluster, but then 80% aren’t and can be speciated with a given gene. So species-resolution is possible with that gene most of the time. This logic of a counter-example meaning species resolution is “impossible” appears in other places in the manuscript as well (and repeatedly in the literature).

Figures are so low-resolution as to be nearly illegible.

**Have the authors made all data and (if applicable) computational code underlying the findings in their manuscript fully available?**

Reviewer #1: **No: **I did not find the link to the code used to generate the analysis

Reviewer #2: Yes

Reviewer #3: Yes

PLOS authors have the option to publish the peer review history of their article (what does this mean?). If published, this will include your full peer review and any attached files.

Reviewer #1: No

Reviewer #2: No

Reviewer #3: No

Figure Files:

Data Requirements:

Re

---

## [Decision Letter · Decision Letter 1]

22 Jul 2024

Dear Dr. Commichaux,

We are pleased to inform you that your manuscript 'Database size positively correlates with the loss of species-level taxonomic resolution for the 16S rRNA and other prokaryotic marker genes' has been provisionally accepted for publication in PLOS Computational Biology.

Best regards,

andre kahles, PhD

Guest Editor

PLOS Computational Biology

Ilya Ioshikhes

Section Editor

PLOS Computational Biology

Reviewer's Responses to Questions

**Comments to the Authors:**

Reviewer #1: The authors have satisfactorily adressed my comments.

Reviewer #2: The authors addressed all my concerns.

Reviewer #3: The authors have provided responses to the questions I previously raised.

**Have the authors made all data and (if applicable) computational code underlying the findings in their manuscript fully available?**

Reviewer #1: Yes

Reviewer #2: Yes

Reviewer #3: Yes

PLOS authors have the option to publish the peer review history of their article (what does this mean?). If published, this will include your full peer review and any attached files.

Reviewer #1: No

Reviewer #2: No

Reviewer #3: No

---

## [Editor Report · Acceptance letter]

31 Jul 2024

PCOMPBIOL-D-24-00016R1 

Database size positively correlates with the loss of species-level taxonomic resolution for the 16S rRNA and other prokaryotic marker genes

Dear Dr Commichaux,

I am pleased to inform you that your manuscript has been formally accepted for publication in PLOS Computational Biology. Your manuscript is now with our production department and you will be notified of the publication date in due course.

With kind regards,

Zsofia Freund
